# Tethering Innate Surface Receptors on Dendritic Cells: A New Avenue for Immune Tolerance Induction?

**DOI:** 10.3390/ijms21155259

**Published:** 2020-07-24

**Authors:** Lucille Lamendour, Nora Deluce-Kakwata-Nkor, Caroline Mouline, Valérie Gouilleux-Gruart, Florence Velge-Roussel

**Affiliations:** GICC EA 7501, Université de Tours, UFR de Médecine, 10 Boulevard Tonnellé, F-37032 Tours, France; lucille.lamendour@gmail.com (L.L.); nora.kakwata-nkor@etu.univ-tours.fr (N.D.-K.-N.); caroline.mouline@univ-tours.fr (C.M.); gouilleux@univ-tours.fr (V.G.-G.)

**Keywords:** dendritic cells, DC subsets, Fc receptors, immune tolerance, pathogen recognition receptors, antibody format, therapeutic antibodies

## Abstract

Dendritic cells (DCs) play a key role in immunity and are highly potent at presenting antigens and orienting the immune response. Depending on the environmental signals, DCs could turn the immune response toward immunity or immune tolerance. Several subsets of DCs have been described, with each expressing various surface receptors and all participating in DC-associated immune functions according to their specific skills. DC subsets could also contribute to the vicious circle of inflammation in immune diseases and establishment of immune tolerance in cancer. They appear to be appropriate targets in the control of inflammatory diseases or regulation of autoimmune responses. For all these reasons, in situ DC targeting with therapeutic antibodies seems to be a suitable way of modulating the entire immune system. At present, the field of antibody-based therapies has mainly been developed in oncology, but it is undergoing remarkable expansion thanks to a wide variety of antibody formats and their related functions. Moreover, current knowledge of DC biology may open new avenues for targeting and modulating the different DC subsets. Based on an update of pathogen recognition receptor expression profiles in human DC subsets, this review evaluates the possibility of inducing tolerant DCs using antibody-based therapeutic agents.

## 1. Introduction

Dendritic cells (DCs) are the most potent professional antigen-presenting cells (APCs). They possess the ability to stimulate CD4^+^ T cells, CD8^+^ T cells, and other immune cells. They represent an essential link between innate and adaptive immunity [1,2]. In mice and humans, DCs are a heterogeneous group that has been broadly subdivided into conventional DCs (cDCs), plasmacytoid DCs (pDCs), Langerhans cells (LCs) and inflammatory DCs (inf-DCs) [3]. All DC subsets have highly effective mechanisms to detect pathogens, virus-infected cells or tumor cells through various receptors, such as pattern recognition receptors (PRRs), Fc receptors (FcRs), complement receptors, and scavenger receptors. On pathogen recognition or damage-associated signaling, DCs undergo maturation and migrate to secondary lymphoid organs to activate effector T lymphocytes (Teffs) and prime adaptive immunity. Depending on the DC subset and type of stimulus, DCs induce either T cell activation or immune tolerance, mainly through regulatory T cell (Treg) induction [4]. However, aberrant activation of DCs is sufficient to cause chronic inflammation, and several studies have shown altered DC function in several patterns of autoimmune diseases [5]. Thus, DCs initiate a protective immune response against infectious diseases, and play pathogenic roles in the onset of autoimmune and inflammatory diseases.

The contribution of DCs to both immunity against pathogens and immune tolerance makes them ideal for inducing either immunity in cancer and infections, or immune tolerance in organ transplantation and autoimmunity. In the last few decades, several studies using adoptive transfer of DC-based anticancer vaccines have been developed [6]. As no therapeutic effect has yet been obtained in humans, in situ stimulation appears to be more effective. Therefore, another approach is required that uses therapeutic antibodies to target immune cells. Very few antibodies have been developed that target DCs, even though they clearly constitute a suitable target for modulating the immune system. This review will provide an overview of some of the most relevant of DC functions, as well as mouse models, which support the therapeutic use of tolerant DCs. We expose the original mechanisms which have been exploited by pathogens to induce tolerant DCs. We will further discuss about the antibody formats that could mimic these strategies, such as bispecific or tri-specific antibodies. Finally, it presents possible therapeutic strategies to induce immune tolerance by modulating DCs using antibodies.

## 2. Dendritic Cell Subsets

Several distinct DC subsets are found in human blood, as well as in most human lymphoid and non-lymphoid tissues. They make up approximately 1% of circulating peripheral blood mononuclear cells and are usually defined as professional APCs that express high levels of major histocompatibility complex class II, MHC II, and co-stimulatory molecules, such as B7.1(CD80) and B7.2 (CD86), but lack other lineage markers, like CD3 (T cell), CD19/CD20 (B cell), and CD56 (natural killer cell). As mentioned before, DCs are subdivided into cDCs, pDCs, LCs, and inf-DCs.

The cDCs are identified by their expression of both MHC II and CD11c. In humans, they are subdivided in two subsets, cDC1 and cDC2, based on their expression of CD141 (BDCA3) and CD1c (BDCA1), respectively (Table 1). Murine cDCs in blood and lymphoid organs are prone to presenting antigens to naive T lymphocytes and can be also subdivided into the two main cDC1 and cDC2 subsets. The classical cDC1 subset is identified by the expression of MHC II, CD11c, and CD8a, or CD103 in skin or non-lymphoid tissues, respectively (Table 1). Like human cDC1, mouse cDC1 expresses XCR1, CADM1, and Clec9A (DNGR-1, CD370) [7]. Similarly, both mouse and human cDC2 express CD11b^high^ and CD4, and mouse cDC2 can be subdivided in two subsets based on the expression of Clec12A and ESAM. 

The pDCs specialize in the production of type I interferons during viral infections, and are critical in antiviral immune response. They are identified by their expression of CD123, CD4, MHCII, CD303 (BDCA2), CD304 (BDCA4), and CD45RA (Table 1) [8]. In mice, pDCs express low levels of CD11c and MHC II, but high levels of B220, Bst2 (CD317), and Siglec-H.

Human LCs are a heterogeneous group that controls the induction of adaptive immune response in the epidermis. They are mainly characterized by the expression of CD207 (langerin), CD1a, E-cadherin, high-affinity IgE receptor (FcεRI), CD39, and Birbeck granules [9]. They have a lower expression of CD11b than mouse LCs and no F4/80 (Table 1). LCs share many features with DCs but their ontology is related with that of tissue-resident macrophages. Unlike cDCs, LCs are largely maintained by self-renewal.

Inf-DCs in mice and monocyte-derived DCs (moDCs) in humans form a distinct population of DCs that appear and differentiate in situ at inflammation sites. They both express MHC II, CD11c, CD1c (BDCA1), CD1a, FcεRI, CD206 (mannose receptor, MR), SIRPα (CD172a), CD14, CD11b, M-CSFR, and CD209 (Table 1) [10]. In steady state conditions, moDCs have been identified in human skin, lung, and intestine tissues [10]. However, during inflammation, LyC6^hi^ monocytes are recruited and differentiated in situ into inf-DCs in mouse models [11]. Mouse inf-DCs exhibit a similar phenotype to cDC2s but with higher expression levels of MHC II, CD11c, and CD11b. They also express Ly6C, F4/80, MR, and FcɛRI. The last of these could be used to discriminate inf-DCs from activated macrophages in addition to CCR7 expression [7].

As summarized in Table 1, each DC subset exhibits a specific PRR pattern which allows it to be characterized and targeted. Even though it is well known that DCs are professional APCs that can induce and orientate the immune response, each DC subset seems to exhibit specific skills concerning the type of T lymphocytes that it can promote, for instance, T helper cells (e.g., Th1, Th2, Th9, Th17) and Tregs. Based on specific properties, such as cross-presentation capacity, and, thanks to knockout mouse models, all these DCs have been shown as highly specialized in their APC functions with few differences between mice and humans. The cDC1s have a great capacity for cross presentation and cytotoxic T lymphocyte induction, while cDC2s are better at promoting Th2 and Th17. In contrast, pDCs outside their anti-viral function seem to be divided between autoimmunity and immune response induction, which illustrates their functional heterogeneity [7]. To sum up, all DCs appear to play a central role in initiating, promoting, sustaining, and controlling the immune response with high-level skills in antigen presentation and the T cell interface. These cells are highly efficient through their capacity to analyze and integrate a vast number of the signals that lead to an immune response or immune tolerance.

## 3. Pattern Recognition Receptors and Fc Receptors on Dendritic Cells

Pathogens, tumor cells, and apoptotic cells present a variety of pathogen- or danger-associated molecular patterns that are recognized by PRRs and FcRs. PRRs are composed of four families: toll-like receptors (TLRs), C-lectin receptors (CLRs), NOD-like receptors, and RIG-I-like receptors [12]. Among the FcR family members, Fcγ receptors (FcγRs) and FcR neonatal (FcRn), which recognizes the Fc structure of immunoglobulin (Ig) G, are those mainly involved in DC activation [13]. TLRs, CLRs, and FcγRs can cooperate to induce immune tolerance or an immune response to eliminate pathogens. PRRs initiate key inflammatory responses and also shape adaptive immunity. As mentioned before, TLRs and CLRs are the most abundant of the PRRs on the DC surface. The two receptor families differ from each other in ligand recognition, signal transduction, and sub-cellular localization. TLRs and CLRs are expressed by myeloid cells, such as DCs, monocytes, and macrophages, as well as by various non-immune cells [12]. FcRs are expressed on the surface of hematopoietic cells that recognize and bind the Fc region of certain Ig classes and subclasses (Table 1). They play crucial roles in antibody-mediated immune responses. Most innate myeloid cells express FcRs for IgG, IgA, and IgE in both mice and humans [14].

Of the 10 TLRs identified in human cells, only TLR1, 2, 4, 5, 6, and 10 are expressed on the surface of the different members of the DC family, whereas TLR3, 7, 8, and 9 are expressed on the endosomal membrane. The TLRs are mainly involved in the induction of inflammatory cytokine and chemokine secretions via the nuclear factor kappa B (NFκB) pathway. Only TLR2 seems to be able to initiate both inflammatory [15] and pro-tolerogenic pathways [16]. TLR2 forms heterodimer complexes with either TLR1, TLR6, or TLR10, and these complexes are critical for the recognition of many diverse microbial structures, including fungal cell walls and lipoproteins from Gram-negative and Gram-positive bacteria, fungi, and viruses [12,17]. Moreover, TLR pattern profiles could also be used to identify each DC subset as some TLRs are expressed on particular DC subsets only. Only pDCs express TLR7 and TLR9; as cDC1s do TLR3; cDC2s, TLR4 and TLR5; and inf-DCs, TLR4 and TLR2 [7] (summarized in Table 1). Mouse DCs have shown several differences in their TLR profile. Mouse and human cDC1s both express TLR3 but mouse cDC1s also express TLR4, TLR11, and TLR13. Similarly, mouse and human cDC2s both express TLR4 and TLR5, but mouse cDC2s also express TLR7, TLR9, TLR11, and TLR13 [7].

CLRs recognize a large and diverse range of ligands (as carbohydrates, lipids, and proteins) in a calcium-dependent manner through the C-type lectin-like domains or carbohydrate recognition domain. They are expressed by all myeloid cells (such as DCs and macrophages) [18], and many isoforms exist for each CLR. CLR family members mediate not only pathogen recognition, but also self- and non-self- antigen uptake, as well as interactions between cells [19,20]. CLRs trigger immune responses by inducing signaling pathways via an immunoreceptor tyrosine-based activation motif (ITAM), ITAM-like motif, or immunoreceptor tyrosine-based inhibitory motif (ITIM) [21]. Activation of ITAM-bearing CLRs mediates the recruitment and activation of tyrosine kinases from the Syk family. Very few CLRs bear the ITIM motif. Examples are DC immunoreceptor (DCIR), Clec12A, Clec12B, and Ly49Q (mouse). However, two CLR groups bear both ITAM/ITIM motif; there are the CLR type II, with dendritic cell-specific intercellular adhesion molecule-3-grabbing non-integrin (DC-SIGN), SIGNR1, LSECtin, MCL, Langerin, and, from the CLR type VI family, MR and DEC205. The situation is more complex with these CLR groups, which can alternate between inflammation and immune tolerance depending on the agonist signal. ITAM-ITIM-independent CLRs do not signal through Syk or phosphatases, although they may contain tyrosine-based motifs involved in endocytosis. More typically, ITIM signaling inhibits ITAM signaling. As an example, DCIR induces inhibition of IL-12 synthesis and production of TNF-α and IFN-γ [22].

Mouse and human FcγRs have received a great deal of attention from the scientific community because of the large expansion of the use of monoclonal antibodies and their involvement in treatment efficacy [14]. In humans, six FcγRs have been identified and subdivided into activating or inhibitory receptors bearing an ITAM or ITIM motif in their intracytoplasmic domain. Activating FcγRs on DCs activated by IgG and immune complexes transduce ITAM signaling via Syk. This leads to co-stimulatory molecule expressions and Th1 activation via IL-12 production. Conversely, inhibitory FcγRs, such as FcγRIIB, inhibit the effects of activating FcγRs on DCs and are involved in immune tolerance to self-antigens from apoptotic cells [14,23]. Mouse FcRs have only four members (FcγRI, FcγRIIB, FcγRIII, and FcγRIV), each with different affinities according to the IgG subclass. Myeloid cell whose DCs express FcγR profiles (this varies between the rest and active states) express a wide variety of FcγRs in both mouse and human cells [24].

The precise FcγR pattern on each DC subset has not yet been fully elucidated. Most studies have been conducted on moDCs in humans, on which only FcγRI (CD64), FcγRIIA (CD32A), and FcγRIIB (CD32B) have been documented [14]. FcγRIIA is present on pDCs, but not FcγRIIB nor FcγRIIC [25]. The receptors FcγRIIA and FcγRIIIA are commonly expressed on both human cDC subsets, whereas FcγRI appears to discriminate the CD1c^+^ subset only [26]. In mice, the FcγR profile seems to be less discriminating than inf-DCs, with CD8^+^, and CD8^-^; DCs present FcγRI, FcγRII, FcγRIII, and FcγRV at various expression levels. Finally, FcγRs on DCs seem not only to influence the Ig interface but also to modulate DC functions [27]. FcRn, which has a different role from the FcγRs, is expressed on all human and mouse DCs and contributes to DC homeostasis [28]. FcRn also seems to be essential to the antigen presentation mechanisms of DCs through immune complex uptake and antigen processing in both human DCs and mouse CD8^-^ DCs [29]. In one way or another, all these receptors have the ability to modulate the immune functions of DCs, but the activation of only a few of them leads to a tolerogenic DC profile.

Nevertheless, because of their central role in regulating the immune response, DCs are also involved in the pathogenesis of a large number of autoimmune diseases, such as rheumatoid arthritis, type 1 diabetes, multiple sclerosis, systemic lupus erythematosus, psoriasis, and allergies [5,30,31]. Altered cytokine secretion by DCs can underlie a deleterious imbalance between Th1, Th2, and Th17, and contribute to pathological mechanisms. Their involvement in pathogenesis clearly justifies their being targeted in situ to moderate or inhibit their inflammatory properties, or even induce tolerant DCs which might contribute to disease control.

## 4. Tolerant Dendritic Cells

Based on current understanding, immune system homeostasis is a dynamic process in which Tregs seem to play one of the main roles. The physiological aim of this homeostasis is to allow a multicellular organism to exhibit self-consistency. The only effective way to induce Tregs is to have in situ tolerant DCs. “Tolerant” DCs (tol-DCs) are defined first and foremost by their function, which is to induce immune tolerance to the antigen encountered [32]. Several mechanisms for inducing immune tolerance have been described so far, including induction of Tregs, anergic or apoptotic T cells, and clonal deletion of T cells. Some tol-DCs are characterized by a state of semi-maturation, the activation of indoleamine-2,3-dioxygenase, the secretion of IL-10 or transforming growth factor (TGF)-β, and the expression of immunosuppressive molecules, such as cytotoxic T lymphocyte antigen 4 (CTLA-4) and programmed-death ligand-1 (PD-L1) [32]. No definitive profile of tol-DCs have been established yet.

Numerous molecules have been tested to induce tol-DCs from both human and mouse DCs. These can downregulate the expression of MHC II and CD80 and CD86 co-stimulatory molecules. The IL-10 production by tol-DCs can also be induced by (i) immunosuppressive drugs, such as corticosteroids (dexamethasone), calcineurin inhibitors (tacrolimus, cyclosporine, and rapamycin), or aspirin; (ii) anti-inflammatory factors, such as 1,25-dihydroxyvitamin D3 (vitD3); and (iii) and TGF-β treatment [33]. In a comparison of vitD3, dexamethasone, TGF-β, rapamycin, and IL-10 as treatment on both immature and mature DCs, Boks et al. clearly demonstrated the superiority of IL-10 in inducing tol-DCs with an effective ability to differentiate Tregs. Moreover, tol-DCs induced by IL-10 treatment express fewer pro-inflammatory cytokines like IL-1β, IL-6, and TNF-α; no IL-12 production was detected. IL-10 treatment also stimulated DC production [34].

Some of these molecules have also been evaluated in mouse models in which disease models could be developed. The treatment of mice with immunosuppressive drugs, such as rapamycin improves islet transplantation [35], skin graft [36], and organ survival [37]. The common feature in all these models is the induction of regulatory lymphocyte populations with various phenotypes and properties. The role of DCs in such induced immune tolerance has been highlighted by adoptive transfers of ex vivo-generated tol-DCs, which resulted in better outcomes in arthritis [38], colitis [39], experimental autoimmune encephalomyelitis [40], and allografts [41]. In grafts, both myeloid and plasmacytoid DCs promote tolerance to allo-antigens [42]. All these models demonstrate the utility of tol-DCs in the treatment of both inflammatory and autoimmune diseases. In fact, clinical trials are in progress in the USA and Europe using adoptive therapy with autologous tol-DCs in organ grafts and autoimmune diseases (nicely reviewed in Reference [43] and Table 2). At least two of the four trials presented in Table 2 have shown clinical improvement in multiple sclerosis [44] and rheumatoid arthritis [45]. The results seem less encouraging in Crohn’s disease even if an increase in Tregs was shown [46]. The last clinical trial was a large project in Europe on kidney graft patients which demonstrated the safety of the treatment, the results of which are eagerly awaited [47]. Nevertheless, applying these strategies in human therapies remains cumbersome, unique to each patient and difficult to implement. In situ targeting of DC subsets would be more powerful in inducing antigen-specific immune tolerance, which might control inflammation in autoimmune diseases and organ rejection in allografts. 

## 5. Pathogen-Induced Immune Tolerance

Although CLRs and TLRs have specific ligands and distinct signaling pathways, CLRs appear to be able to influence the immune response due to TLRs [48,49]. This in vivo dynamic cooperation between TLRs and CLRs on the DC surface seems to be used to maximize the immune response and to control inflammation. Dectin-1 can modulate the signaling of other TLRs [50]. As an example, Eberle et al. showed that co-stimulation of Dectin-1 and TLR9 induced a downregulation of IL-12p40 and NFκB inhibition in a murine model [51,52]. Furthermore, TLR2, TLR4, and TLR9 recognize ligands from *Mycobacterium tuberculosis*, leading to a simultaneous and/or sequential activation of TLR pathways to eradicate the pathogen [53]. Other receptors, such as DC-SIGN, indirectly increase pro-inflammatory cytokine production by acting in cooperation with TLRs [54,55]. Geijtenbeek et al. showed that the interaction of DC-SIGN with pathogens having mannose structures modulated the TLR4 signaling pathway [56]. This interaction depended on NFκB activation by TLRs and was not limited to TLR4 [57]. Studies have reported that DCIR decreased TLR8-mediated inflammatory cytokine production in cDCs [58] and TLR9-mediated type I IFN in pDCs [22].

This cooperation mechanism has been also described between FcγRs and CLRs/TLRs. FcγRs modulate PRR signaling in DCs either directly through ITAM-like motifs or indirectly through the ITAM-containing adaptor DAP-12 or FcRγ. FcRγ has also been described in murine models as a negative regulator of Dectin-1 signaling due to the interaction between Dectin-1 and other FcRγ transduction receptors [59]. In humans, FcγRIIA has a synergistic effect on moDCs with endosomal TLRs (e.g., TLR3, TLR7, and TLR8) by increasing cytokine production [60]. This cooperation may depend on MyD88 and TRIF signaling after TLR activation.

Moreover, many pathogens are capable of diverting the system to inhibit or modulate the immune response using these pathways [61]. Engagement of DC-SIGN by pathogens with mannose structures, like *Candida albicans*, human immunodeficiency virus-1, and *M. tuberculosis*, modulates TLR4 [62] and TLR2 signaling pathways in particular. *M. tuberculosis* and *M. bovis* are also able to produce mannose-capped lipoarabinomannan (ManLAM), a glycolipid targeting DC-SIGN that interferes with TLR2 signaling, thereby inducing IL-10 secretion by DCs [62,63]. In this context, DC-SIGN activation by ManLAM also modulates the TLR4-mediated NFκB signaling pathway and modifies the IL-10/IL-12p70 secretion balance in favor of IL-10 to promote the infection. This study demonstrates that TLR2, TLR4, and DC-SIGN interactions on the DC surface could have important implications in the course of infections. Dectin-1 can also modulate TLR2 signaling pathways. Zymosan, a cell wall extract from *Saccharomyces cerevisiae*, is recognized by both Dectin-1 and TLR2, thus enabling immune tolerance induction through IL-10 regulatory cytokine secretion [16]. Cooperation between TLR2 and Dectin-1 involves a physical link between these two receptors, suggesting a modulation of the signaling pathway after receptor activation [64].

Through numerous pathways using this kind of cooperation between PRRs and FcγRs, pathogens have been able to induce local or systemic and transient or permanent immune tolerance leading to immune system subversion. Cooperation between TLRs, CLRs, and FcγRs can prevent excessive immune activation, thereby either attenuating pro-inflammatory signals or enabling pathogens to evade the immune system. Inhibition of the immune system is induced by anti-inflammatory cytokines, such as IL-10, TGF-β, IL-13, and IL-35, as well as by the internalization and degradation of PRRs [65].

## 6. Antibodies for Tolerant Dendritic Cells

Drug treatments are not specific enough to target DCs in tol-DC induction. Thus, new approaches, such as antibodies targeting DC surface receptors, are being developed. As immune tolerance can be induced by selective delivery of antigens to immature DCs, a chimeric antibody specific for the CLR DEC205 (CD205) has been developed [66]. The activation of DEC205 on the DC surface activated the endocytic pathway and thus antigen processing and presentation on the DC surface. Targeting DEC205 with the chimeric antibody coupled with an antigen, such as ovalbumin (OVA) or myelin oligodendrocyte glycoprotein, led to Treg differentiation and induced anergic T cells. The Tregs induced by anti-DEC205 treatment expressed CD4, CD25, FoxP3, and secreted IL-10 [67]. In contrast, targeting murine CD11c^+^ DCs with antibodies against langerin bound to OVA drives a CD4^+^ and CD8^+^ specific response [68]. Antibodies against DCIR that target different DC subsets in both mouse and human models have led to an effective, specific response [69].

Tol-DCs are an attractive treatment alternative for inducing a tolerant environment to limit organ graft rejection or autoimmune pathologies [33]. Therefore, therapeutic antibodies (Abs) targeting DCs to induce immune tolerance may represent a promising strategy for developing new therapies to treat all immune disorders [70]. To that end, one might imagine a new antibody format that can fuse certain PRRs together by mimicking the subversive strategy of pathogens and modulate DC functions to induce tol-DCs.

Molecular engineering makes a large number of Ab formats possible, from Abs variable heavy-chain Abs (VHH) to whole Abs (Figure 1). Indeed, from VHH, single-chain variable-fragments (scFvs), bispecific T-cell engagers, and diabodies to larger molecules (e.g., 250-kDa IgG-2scFv), all formats have been reviewed [71] and nicely summarized [72,73]. Other approved formats have emerged in the Food and Drug Administration (FDA) or European Union Medicine Agency (EMA), such as bispecific Abs (BsAbs), blinatumomab in cancer, emicizumab in hemophilia, or nanobodies, like caplacizumab, in hemostasis, offering new functional possibilities [74]. Bispecific formats with VHHs or scFvs in tandem (Figure 1a,b) seem to be the most appropriate Abs for obtaining TLR and PRR crosslinking. However, their half-life is not compatible with clinical use. This issue could be remedied by adding an Fc domain, although this may pose other problems, such as interference with the 3D shape of the BsAb and space length between the two targets. Nevertheless, some functional examples do exist, including a BsAb against P2X7 and a specific cell marker that combines two different VHHs and an Fc domain (Figure 1c). These BsAb formats are being seriously evaluated in inflammatory bowel disease [75] and asthma [76]. Many novel BsAbs are now in clinical development or the late pre-clinical phase [77].

Concerning trispecific Abs (TsAbs), a clear definition must first be established as some “TsAbs” are only tri-functional Abs that combine two binding paratopes and an Fc part with complement-dependent and antibody-dependent cell-mediated cytotoxicity activity. For us, TsAbs are molecules which bind three epitopes together. Effective TsAbs have been developed in infectious diseases [78] and cancer [79,80]. There are many formats for building TsAbs, from scFvs and VHH to Fab fusions [81]. Wu et al. have described orthoTsAbs binding PD-1, CD137, and CTLA4 molecules based on Fab fusions [79]. Patents already exist for several TsAbs, including one TsAb developed by Adimab LLC (WO2018/148445) (Figure 1d), a combination of three heavy and light chain variables (VH/VL) with an Fc domain (Figure 1e) [82], or a VH/VL combination for a TsAb against CD30/CD19/CD16A developed by Affimed Therapeutics (WO2017/064221) (Figure 1f). These molecules are only at the patent stage and their characteristics and functional properties remain to be studied. TsAbs designed to combine TLRs, PRRs, and FcγRs appear trickier, as there are considerable constraints with regard to epitope accessibility, distance control between partners, and activation of biased transduction, characteristics that are in fact expected from these Abs. Still, it may be supposed that all kinds of formats will appear in the coming years. The field of Ab formats is boundless, its one limit is the producibility of the molecules. We assume that tri-, quadri-, or multi-specific Abs must be now envisaged to meet the needs that will arise from the new immunological mechanisms being explored.

## 7. Concluding Remarks

Based on all that we have seen in this review, the targeting of CLRs, TLRs, or FcγRs might enhance the specific targeting of DC subsets, and the tethering of several surface receptors might screw the DC balance to immune tolerance. Moreover, therapies using multi-specific Ab formats will provide a more potent way of in situ modulating DC functions and inducing tol-DCs which could become an appropriate alternative therapeutic strategy in immune diseases.

## Figures and Tables

**Figure 1 ijms-21-05259-f001:**
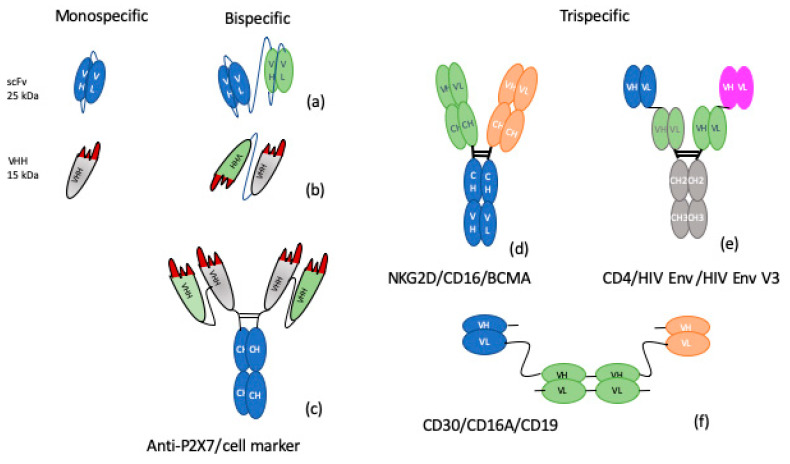
Alternative formats for bispecific or trispecific antibodies. From left to right: Monospecific, VH, VL, and VHH, (nanobody); Bispecific, antibodies, such as scDiabody (**a**), Diabody (**b**), and tandem VHH, on Fc (**c**). Trispecific, formats are represented as combinations of three VH/VH domains fused via six CH domains (**d**), three different VH/VL domains fused to Fc (**e**), and three VH/VL domains with linkers (**f**). VL, VH, and VHH domains are shown in blue, green, gray, and orange according to their respective antigen specificity. Peptide linkers are shown as thin black lines. with VH = variable domain of the heavy chain; VL = variable domain of the light chain; VHH = variable heavy Homodimer, CH = constant domain of heavy chain.

**Table 1 ijms-21-05259-t001:** Phenotypes and functions of dendritic cells (DC) subsets.

Subsets	Human					Mouse				
	pDC	cDC1	cDC2	LC	mo-DC	pDC	cDC1/CD8^+^	cDC2	LC	Inf-DCs
**Location**	Blood, lymphoid tissues	Blood, lymphoid tissues	Blood, lymphoid tissues	Skin	Inflamed tissues	Blood and lymphoid tissues	Blood and lymphoid tissues	Blood and lymphoid tissues	skin	Inflamed tissues
**Phenotype**	Lin^-^, CD123, CD45RA, CD304 CCR9+/−	Lin^-^, CD141, XCR1, CD11c^low^	Lin^-^, CD1c, CD11c^high^, SIRPα, CD26	CD11c^low^, CD32, CD1a, CD1c, CD123, E-cadherin, EpCAM, SIRPα	Lin^-^, CD11c, CD1a, CD1c, CD141,	Lin^-^, CD11c^int^, LyC6, Ly-49Q, CD45RAB220, SCA-1, CD9, CCR9, Bst-2	Lin^-^, CD8, CD11c, CD24, (CD103^+^)ICAM, XCR1	Lin^-^, CD4, CD11b^high^, CD11c^high^, CD26, SIRPα (CD103^+)^, Clec12A/ESAM	CD11b, CD1a, CD24 F4/80, SIRP, epCAM	Lin^-^, Ly6C, F4/80 CD11b^high^, MHC-II^high^, CD11c^int^
**PRRs and FcγRs expression**	TLR6,7,9,10* DCIR, CD303 Dectin-2, FcγRIIA^low^, FcRn	TLR1,3,6,8,10 * Dectin-1, 2 CD209, Clec9A, CD370, DCIR FcγRIIA^int^, FcRn, FcγRIIB	TLR1,2,4,5,6,8,10 Dectin-1,2 CD209^low^, CD206, CD301DCIR, CD370, Clec10A FcγRI, FcRn FcR^high^ FcγRIIB FcγRIIIA^low^	TLR2,3,4,6, CD207^high^, CD205, FcγRIIB, FcRn	TLR2,4, CD209, CD206, DCIR, CCR7 FcγRI ^low^ FcγRIIA, FcγRIIB FcRn, *F*γ*cRIIC*	TLR2,7,9, Siglec-H, DCIR2, FcγRIIB FcγRIIIB	TLR3,4,11,13 CD205, Clec9A, DNGR1, FcγRI, FcγRIII, FcγRIV, (FcγRIIB)	TLR2,4,5,7,9, 11,12,13, DCIR2, dectin1 Clec12A FcγRI, FcγRIIB, FcγRIII, FcγRIV	TLR7,8 CD207, CD205, CD301b, CD370, FcγRIIB,	TLR2,4,6,8,9 CD209, CD205, CD206, CCR2 FcγRI, FcγRIII, *Fc*γ*RIIb*
**Functions**	Viral and cancer response, Th1, Treg	Th1, CTL Priming, Cross P.	Th2, Th1, Th17 Tolerance Cross P.	Th2, CTL response, Tolerance	Naïve and memory Th, Th1 Th2, Th17, tol-DCs	Th1, Th17, Treg Cross. P ^low^	Th1, Treg, Cross P. (CD207^+^)	Th2, Th17, CD4^+^ priming	Th2, Tolerance Cross P.	Th1, Th17 Tolerance, Cross P.

Lin^-^: CD3, CD19, CD56; cross P., cross presentation; * Immgene data; *only in inflammation;* () in tissue.

**Table 2 ijms-21-05259-t002:** Clinical trials involving tol-DCs.

Immune Disorder	Phase	Tol-DC Generation	Antigen	Status	Major Outcomes	N°	Reference
Multiple sclerosis	1	Vit D3 Dex, mo-DCs	myelin peptides myelin peptides	Active/recruiting completed	Ongoing studies	NCT02618902 NCT02283671	[44]
Rheumatoid arthritis	1	With Dex, vit D3, and monophosphoryl lipid A, mo-DCs	Synovial fluid	Active/recruiting	Clinical improvement only for high doses (3-10.10^6^ tol-DCs)	NCT01352858	[45]
Crohn Disease	1	Vit A, Dex mo-DCs	No antigen	Completed	Clinical improvement was observed in 33% of the patients and increase of circulating Tregs and decrease in IFN-γ levels.	NCT02622763	[46]
Organ graft (kidney)	1,2	Autologous tol-DCs (mo-DCs) from live donors	No antigen	Completed	Not yet published	NCT02252055	[47]

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
