# Peer review of "Tethering Innate Surface Receptors on Dendritic Cells: A New Avenue for Immune Tolerance Induction?"

_ijms, 2020, doi:10.3390/ijms21155259_

Round 1

Reviewer 1 Report

An Interesting and well written review article with current information of the function and phenotypes of various dendritic cell subsets in human and mouse systems, and the pattern recognition and Fc receptors expressed on that cell populations, pointing out the aspects of tolerogenesis mediated by dendritic cells (DCs) in different immune diseases and their pathogenic aspects of tolerance in infections. Authors discuss genetic modified immunoglobulin forms as a tools for tolerance activation involving DCs as a new proposition of modulation their function.

 Minor revision:

Since IgG4 subset of mouse immunoglobulin, as an exemption, does not express the activity of binding FcgR, could you explain more critically the sentence from the line 87-88.

Please use the form of double capital letter in case of interleukin abbreviations:  IL instead of Il see lines 79, 119. Please write together the name of IL-10 – line 130-131

Please delete the bracket in line 63 and wrote together CD86 instead of (CD) 86

In Abbreviations: please use the Greek letters in  case of TNF-alfa and TGF-beta

Author Response

Minor revision:

Since IgG4 subset of mouse immunoglobulin, as an exemption, does not express the activity of binding FcgR, could you explain more critically the sentence from the line 87-88.

Line 91 - We agreed and added in the sentence the following information: “except for FcgRIV which has no binding properties on FcγRs.”

Please use the form of double capital letter in case of interleukin abbreviations:  IL instead of Il see lines 79, 119. Please write together the name of IL-10 – line 130-131

Line 81,125- We changed all miswriting of Interleukin (IL).

Please delete the bracket in line 63 and wrote together CD86 instead of (CD) 86

Line 61,63 - We thank the reviewer of his deep proofreading and corrected the text.

In Abbreviations: please use the Greek letters in  case of TNF-alfa and TGF-beta

Line 141,142- We changed the two letters by their Greek equivalents.

Reviewer 2 Report

The review paper entitled: "Tethering Innate Surface Receptors on Dendritic 2 Cells: A New Avenue for Immune Tolerance 3 Induction?" summarized the scientific discoveries about DCs and proposed an interesting perspective of using tandem antibodies to induce tol-DCs. The manuscript is well-written with a clear structure for the readers to understand. Only some minor questions may need to be considered in the revision:

(1) What are the specific markers on the Tol-DCs compared to the cDCs? How to identify Tol-DCs from the periphery? Does progenitor Tol-DC exist?

(2) Are the novel antibodies used to induce tol-DCs targeting DCs in the PBMCs? LNs, Spleen, skin, and many other organs have more DC populations compared to the blood. Is there some approach to facilitate the antibodies to target DCs in these tissues?

(3) The sections 1,2,3 are very informative and well-written. But consider shortening a little to emphasize the tolerant DC content which is the most important in this review.

Author Response

Comments and Suggestions for Authors

The review paper entitled: "Tethering Innate Surface Receptors on Dendritic 2 Cells: A New Avenue for Immune Tolerance 3 Induction?" summarized the scientific discoveries about DCs and proposed an interesting perspective of using tandem antibodies to induce tol-DCs. The manuscript is well-written with a clear structure for the readers to understand. Only some minor questions may need to be considered in the revision:

(1) What are the specific markers on the Tol-DCs compared to the cDCs? How to identify Tol-DCs from the periphery? Does progenitor Tol-DC exist?

- As far as the bibliography reports, no specific progenitors have been described for tol-DCs. The capacity to become tolerant seems possible for all DCs according the environmental conditions, even though mo-DCs seem to be one of DCs that become tol-DCs most easily. As for Treg, the associated expression of different markers allows the tol-DCs identification. The markers which can to used are IL-10, TGF-b Receptors, PD-L1, IL-T3, IL-T4, ICOS-L, GAL-3 (galectin-3), IDO, …no exhaustive list is available as written on line 127.

 (2) Are the novel antibodies used to induce tol-DCs targeting DCs in the PBMCs? LNs, Spleen, skin, and many other organs have more DC populations compared to the blood. Is there some approach to facilitate the antibodies to target DCs in these tissues?

Today, no antibody have been developed targeting DCs. In fact, very few DCs are circulating and the highest concentrations of DCs will be founded in lymphoid organs. Lehmann (Lehmann et al., 2016) suggested that antibodies against surface molecular markers of DCs will be helpful and we think that bi or tri-specific antibodies might improve the DC targeting.

(3) The sections 1,2,3 are very informative and well-written. But consider shortening a little to emphasize the tolerant DC content which is the most important in this review.

We agreed and would like to say that very relevant strategies to induce tol-DC with product of potential therapeutic are not so numerous. Products such as VitD3, 14-DHE will not have therapeutic development in human shortly. We have presented the examples from literature which appear having a therapeutic potential to us.

All paragraphs are about the same number of words except the DCs section which reports the complexity of the “DCs” world in term of  type of subsets, markers and properties.

Reviewer 3 Report

In this review Lamendour and cols. describe the present classification of DCs and summarize the current knowledge on the distribution and function of pattern recognition receptors and Fc receptors on dendritic cells especially in their importance to generate tolerance.

The review overall is well structured, easy to read and interesting. Authors provide the reader with general background on dendritic cells subsets, pattern recognition receptors and Fc receptors distribution and function on dendritic cells. They describe the current knowledge about tolerant dendritic cells and the influence of pathogens and antibodies in this DC tolerance.

In general the manuscript is interesting and well written but my major concern is related to the fact that the title point at the use of antibodies to the induction of tolerance in DCs but this is a small part in the manuscript. I suggest to try to expand a little more this novel and very important part of the manuscript which make it different from other reviews.

Minor concerns

Author list: The surname of the authors are in capital letters

Line 70. The classical cDC1 subset is identified by the expression of MHCII, CD11c, and CD8a, or  CD103 in skin or non-lymphoid tissues, RESPECTIVELY (Table 1). Maybe a “respectively” could clarify the sentence even more.

Line 84 there is a double space between “activated” and “by”

Line 121. A reference should be included after “and programmed-death ligand-1 (PD-L1).” Maybe the reference 32 could be used here.

Line 93 (Second numbering) a space should be removed between “/” and “CD16A” in CD30/CD19/ CD16A

Figure 1. The lettering of the figure is difficult to read in the boxes, maybe an increased size of the letters and a higher resolution of the figure may allow a better reading.

Bibliography. The bibliography should be format to IJMS style

Author Response

Minor concerns

Author list: The surname of the authors are in capital letters

-We made the names in line with IJMS format.

Line 70. The classical cDC1 subset is identified by the expression of MHCII, CD11c, and CD8a, or  CD103 in skin or non-lymphoid tissues, RESPECTIVELY (Table 1). Maybe a “respectively” could clarify the sentence even more.

-We had the term “respectively” in sentence line 70.

Line 84 there is a double space between “activated” and “by”

-Authors tanks the reviewer for his serious and deep proofreading of the manuscript. We removed the inappropriate double spaces.

Line 121. A reference should be included after “and programmed-death ligand-1 (PD-L1).” Maybe the reference 32 could be used here.

-We add the reference Steinmann et al., (32)

Line 93 (Second numbering) a space should be removed between “/” and “CD16A” in CD30/CD19/ CD16A

Line 93-Authors tanks the reviewer for his serious and deep proofreading of the manuscript. We removed all the inappropriate double spaces.

Figure 1. The lettering of the figure is difficult to read in the boxes, maybe an increased size of the letters and a higher resolution of the figure may allow a better reading.

We increased the font size of graphs up to 14 (Calibri).

Bibliography. The bibliography should be format to IJMS style

We made the bibliography in line with IJMS format.